# Improving Generative and Discriminative Modelling Performance by Implementing Learning Constraints in Encapsulated Variational Autoencoders

**Wenjun Bai** , **Changqin Quan \* and Zhi-Wei Luo**

School of System Informatics, Kobe University, 1-1, Rokkodai-cho, Nada-ku, Kobe 657-8501, Japan; zokbwj@gmail.com (W.B.); luo@gold.kobe-u.ac.jp (Z.-W.L.)

**\*** Correspondence: quanchqin@gold.kobe-u.ac.jp

**Featured Application: Implementing learning constraints on our introduced encapsulated variational autoencoders permits us to learn representations that can be utilised in tackling discriminative and generative tasks. This approach can be applied to medical-image analysis where the volume of unlabelled data instances, e.g., medical images such as fMRI scanning from certain patients, is disproportionately larger than annotated instances.**

**Abstract:** Learning latent representations of observed data that can favour both discriminative and generative tasks remains a challenging task in artificial-intelligence (AI) research. Previous attempts that ranged from the convex binding of discriminative and generative models to the semisupervised learning paradigm could hardly yield optimal performance on both generative and discriminative tasks. To this end, in this research, we harness the power of two neuroscience-inspired learning constraints, that is, dependence minimisation and regularisation constraints, to improve generative and discriminative modelling performance of a deep generative model. To demonstrate the usage of these learning constraints, we introduce a novel deep generative model: encapsulated variational autoencoders (EVAEs) to stack two different variational autoencoders together with their learning algorithm. Using the MNIST digits dataset as a demonstration, the generative modelling performance of EVAEs was improved with the imposed dependence-minimisation constraint, encouraging our derived deep generative model to produce various patterns of MNIST-like digits. Using CIFAR-10(4K) as an example, a semisupervised EVAE with an imposed regularisation learning constraint was able to achieve competitive discriminative performance on the classification benchmark, even in the face of state-of-the-art semisupervised learning approaches.

**Keywords:** deep generative model; learning constraint; representation learning

---

## 1. Introduction

### 1.1. Deep Generative Models

Representation learning, a learning process that aims to extract representative latent representations of observed data, is one of the most active research areas in artificial intelligence [1,2]. Regarding different task demands, learned representations can coarsely be divided into two major categories: generative and discriminative representations.

Relying on a data-generating latent space, generative models are able to recover salient generative representations of such data that excel in lossless input reconstruction. Utilising deep neural networks for parameter estimation, deep generative models grant further flexibility in learning generative representations that fuels the optimisation of data-reconstruction fidelity [3]. Two leading deep

generative models are variational autoencoders (VAEs) [4] and generative adversarial networks (GANs) [5]. As GAN, the latter approach, is difficult to train and implicitly infer data distribution, we chose VAE as our default deep generative model in this research.

To improve the generative modelling performance of a deep generative model, numerous approaches have been introduced, ranging from the inclusion of the autoregressive principle [6] to mutual information [7]. These refinements allow a deep generative model to learn disentangled latent representations of data to grant further flexibility in data generation. However, while these learned latent representations have fuelled many exciting downstream applications, deep generative models generally perform poorly on discriminative tasks, even with sufficient added supervised signals [8,9]. A simple remedy is to train deep generative models with semisupervised training datasets. This leads to the coinage of deep conditional generative models. However, in comparison to other nongenerative semisupervised learning algorithms, deep conditional generative models are less competitive in terms of their discriminative performance.

As a result, while some generative approaches have previously been investigated, a more efficient approach to improve both the generative- and discriminative-modelling performance of a deep generative model is still lacking.

*1.2. Approach Overview*

To close the foregoing gap, in this research we draw inspiration on coding efficiency from human visual processing. The success of early visual processing in vivo relies on two learning constraints: dependence-minimisation [10] and regularisation [11] constraints. The dependence-minimisation learning constraint encourages internal representations to be diversified, whereas the regularisation constraint stabilises these generated (encoded) internal representations for visual-concept acquisition.

However, how to implement the prior-mentioned biological learning constraints in a deep generative model remains unclear. To address this, we introduce a novel deep-generative-model form: encapsulated variational autoencoders (EVAEs) that stack two different-component VAEs in a scaffold. To implement learning constraints on an EVAE, we introduce a key tunable hyperparameter $\alpha$ to tune the relation between two component VAEs within an EVAE. Thus, the dependence-minimisation constraint, which enforces this derived generative model to encode various latent representations, can be implemented by tuning this $\alpha$ to a large value, whereas a small $\alpha$ can be viewed as the implementation of the regularisation learning constraint to elevate the discriminative modelling performance of a semisupervised EVAE.

The rest of the article is organised as follows. Prior to the description of our EVAE, we outline an economical review on previous attempts to improve the generative and discriminative modelling performance of deep generative models. A detailed description of our EVAE is then presented. Two neuroscience-inspired learning constraints were further delineated, along with the learning algorithm for the EVAE. Empirically, using the MNIST dataset as an example, we show that the imposed dependence-minimisation constraint elevates EVAE generative modelling performance, allowing the model to generate MNIST-like digits in varied patterns of generation. To assess the role of the regularisation constraint on a discriminative task, we started by deriving a semisupervised EVAE model to show that, by imposing the regularisation constraint on a semisupervised EVAE, it achieves competitive discriminative performance, even in the face of state-of-the-art semisupervised learning algorithms on the CIFAR-10(4K) classification benchmark.

## 2. Related Works

In order to improve the generative performance of a deep generative model, early attempts were the proposal of restricted Boltzman machines [12] and the usage of bilinear models [13]. Recent contributions focus on deriving more convoluted generative loss functions, e.g., in a GAN-based generative model by fuelling the original loss function of a GAN with a term to maximise mutual information between latent variables and observed data; InfoGAN [7] achieved spectacular results

in recovering disentangled latent representation. In the same vein, Beta-VAE [14] also excelled in learning disentangled latent representations by adding a disentanglement metric on the original VAE learning objective.

To improve the discriminative performance of a deep generative model, numerous efforts were made, from the direct employment of label information to the semisupervised learning of a deep generative model. The former approach includes the direct employment of label pairing [15] and supervision grouping [16]. The semisupervised learning paradigm, on the other hand, aims to utilise generative (unsupervised) trained representations to regularise the discriminative trained representations to improve their performance in classification tasks. These attempts range from the early convex combination of discriminative and generative classifiers [17] and the usage of the kernel method [18] to recent ladder networks [19], the spike-and-slab sparse-coding approach (S3C) [20] and the virtual adversarial training approach [21].

Unfortunately, these approaches are specifically tailored for improving either generative or discriminative modelling performance of a deep generative model. In comparison with the above-mentioned works, our proposed approach yields a much broader research scope, as it allows the learning of high-quality generative and discriminative representations by simply imposing different learning constraints on the same model.

## 3. Encapsulated Variational Autoencoders

### 3.1. Brief VAE Review

Based on coding theory [22], a VAE can be perceived as a pair of probabilistic encoders, an approximated posterior and a probabilistic decoder, which corresponds to a specified probabilistic model with a defined prior on the decoder.

To formalise our discussion on VAEs, we considered the depicted graphic model in Figure 1A, where $x$ and $z$ correspond to observable and latent variables, respectively. Assuming some dataset $X = \{x\}_{i=1}^{N}$ with $N$ i.i.d. samples of observable variables $x$ and unobservable (latent) continuous random variables $z$, the probabilistic decoder (the model likelihood) can be defined as $p_\theta(x|z)$, with a standard normal prior over the latent variables as $p(z) = N(z|\mu, \sigma)$.

To amortise the inference and approximate the intractable posterior, i.e., $p_\theta(z|x)$, a VAE introduces a variational encoder $q_\phi(z|x)$ to approximate the intractable one, i.e., $q(z|x; \phi) \approx p(z|x; \theta)$, where $\phi$ are to-be-optimised local variational parameters [23].

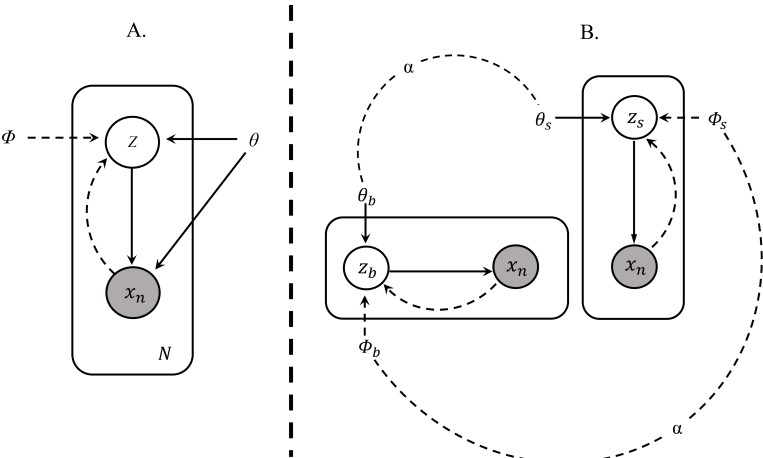

**Figure 1.** Variational autoencoder (VAE) vs. encapsulated VAE (EVAE). In this figure, we present the graphic model representations of (**A**) a VAE and (**B**) the derived EVAE. Solid arrows indicate decoders, whereas dotted arrows represent encoders.

### *3.2. EVAE*

Different from a conventional VAE (cf. Figure 1A), in our proposed EVAE (cf. Figure 1B), the employment of two latent variables under coding theory implies a structured novel encoder–decoder architecture, i.e., two separate probabilistic encoders, e.g., $q_{\phi_b}(z_b|x)$ and $q_{\phi_s}(z_s|x)$, and decoders, e.g., $p_{\theta_b}(\tilde{x}|z_b)$ and $p_{\theta_s}(\tilde{x}|z_s)$ that compile up to two VAEs, i.e., denoting as $VAE_b$ and $VAE_s$. Crucially, despite receiving identical input, the two VAEs differed in terms of their network architectures. This allowed learning two different sets of parameters, e.g., $\{\theta_b; \phi_b\}$ and $\{\theta_s, \phi_s\}$ for $VAE_b$ and $VAE_s$, respectively.

Representation-learning and data-generative processes can be described as two latent representations ($z_b$ and $z_s$) that are first encoded from probabilistic encoders; then, these latent representations are fed to the probabilistic decoders to reconstruct data instances.

Importantly, grouping two latent variables together, e.g., $(z_b, z_s)$, allows us to derive the joint encoder and decoder, i.e., $q_{\phi_b,\phi_s}((z_b, z_s)|x)$ and $p_{\theta_b,\theta_s}(\tilde{x}|(z_b, z_s))$. However, it is essential to note here that we do not assume the conditional independency of two latent variables, i.e., factorised joint encoder $q_{\phi_b,\phi_s}((z_b, z_s)|x) = q_{\phi_b}(z_b|x) \cdot q_{\phi_s}(z_s|x)$. The factorised joint decoder does not hold in this research.

### *3.3. Parameterisations on EVAE*

Prior to demonstrating the critical learning constraints on this derived EVAE, we first introduce the parameterisations that are used in the component encoders, $q_{\phi_b}(z_b|x)$ and $q_{\phi_s}(z_s|x)$. and decoders, $p_{\theta_b}(\tilde{x}|z_b)$ and $p_{\theta_s}(\tilde{x}|z_s)$.

In this case, we parameterised base encoder $q_{\phi_b}(z_b|x)$ under a simplified multivariate Gaussian distribution as $\log q_{\phi_b}(z_b|x) = \log \mathcal{N}(z_b; \mu_b, \sigma^2)$, where the optimised variational parameters are $\phi_b$, which can be used to produce the mean ($\mu_b$) and s.d. ($\sigma^2$) of the approximating distribution.

Different from the simplified parameterisation that is used in the base encoder, we let more complex full-rank Gaussian distribution to parameterise the scaffolding encoder as follows: $\log q_{\phi_s}(z_s|x) = \log \mathcal{N}(z_s; \mu_s, L)$.

Where parameterised variational parameter $\phi_s$ is the concatenation of the mean vector and decomposed covariance matrix $\{\mu_s, L\}$. Here, we used numerical stable Cholesky decomposition to decompose correlation matrix $\Sigma$, into two lower triangular matrices, $\Sigma = LL^T$, to speed up variational inference.

Armed with encoded latent representations ($z_b$ and $z_s$), we let two decoders, $p_{\theta_b}(\tilde{x}|z_b)$ and $p_{\theta_s}(\tilde{x}|z_s)$, take similar forms of multivariate Gaussian that are used in forming the preceding encoders: $\log p_{\theta_b}(\tilde{x}|z_b) = \log \mathcal{N}(x; \mu_b, \sigma^2)$ and $\log p_{\theta_s}(\tilde{x}|z_s) = \log \mathcal{N}(x; \mu_s, L)$.

## 4. Learning Constraints in EVAE

### *4.1. Joint Encoder and Decoder*

To implement the learning constraints in EVAE, we derived the analytic forms of the joint encoder and decoder of the EVAE. The making of the joint encoder and decoder should satisfy the two following requirements: (1) analytic expressions should be flexible enough to include earning constraints, and (2) derived expressions need to be fully differentiable to enable the fast and accurate approximate marginal inference of variable $x$.

To this end, we considered the analytic expression of our joint encoder that allowed the smooth interpolation of the two component encoders in Equation (1):

$$q((z_b, z_s)|x; \phi_b, \phi_s, \alpha) \propto q_{\phi_b}(z_b|x) \cdot q_{\phi_s}(z_s|x)\alpha \cdot \{\mathcal{R}_e(z_b, z_s)\}. \tag{1}$$

Note that, here, we inserted a hyperparameter $\alpha$ to exert direct control over the output of the discrepancy function, $\mathcal{R}_e(z_b^l, z_s^l)$. Importantly, the range of $\alpha$ is defined as $(0, +\infty)$ in this research. This prevents the proposed EVAE from degrading to two unrelated VAEs (the $\alpha = 0$ case).

The introduced discrepancy function $\mathcal{R}_e(z_b, z_s)$ can be further expressed in the following Equation (2):

$$\mathcal{R}_e(z_b, z_s) = \frac{1}{L} \sum_{l=1}^{L} \exp\{-||z_b^l - z_s^l||^2\}. \tag{2}$$

As the samples are easier to work with, we further Monte Carlo-sampled these encoded representations from two encoders, $z_b^l \sim q(z_b|x)$ and $z_s^l \sim q(z_s|x)$, where $l$ stands for the number of samples. The added exp function was to ensure that the measured representation differences were positive.

Similar to the prior factorisation in the encoder case, the analytic expression of the joint decoder is $p_{\theta_b, \theta_s}(\tilde{x}|(z_b, z_s))$, obtaining a similar form in Equation (3), as follows:

$$p_{\theta_b, \theta_s}(x|(z_b, z_s); \alpha) \propto p_{\theta_b}(\tilde{x}_s|z_b) \cdot p_{\theta_s}(\tilde{x}_s|z_s)\alpha \cdot \{\mathcal{R}_d(\tilde{x}_b, \tilde{x}_s)\}. \tag{3}$$

Note here that we also incorporated the same hyperparameter $\alpha$ to scale the differences of the two reconstructed data instances, where the discrepancy function for the joint decoder $\mathcal{R}_d(\tilde{x}_b, \tilde{x}_s)$ can be defined in Equation (4):

$$\mathcal{R}_d(\tilde{x}_b, \tilde{x}_s) = \frac{1}{L} \sum_{l=1}^{L} \exp\{-||\tilde{x}_b - \tilde{x}_s||^2\}. \tag{4}$$

Here, $\tilde{x}_b, \tilde{x}_s$ represent the reconstructed data instances from $VAE_b$ and $VAE_s$, respectively, i.e., $\tilde{x}_b \sim p_{\theta_b}(\tilde{x}|z_b)$ and $\tilde{x}_s \sim p_{\theta_s}(\tilde{x}|z_s)$.

## 4.2. Learning Objective

The model is complete by defining a simple factorised joint prior on two latent variables in Equation (5):

$$p_{\theta_b, \theta_s}(z_b, z_s) = p_{\theta_b}(z_b) \cdot p_{\theta_s}(z_s). \tag{5}$$

Armed with the defined joint encoder (cf. Equation (1)), joint decoder (cf. Equation (3)), and the joint prior (cf. Equation (5)), we could finally derive the objective function of our proposed EVAE.

Recall the learning objective of the original VAE in Equation (6) [4],

$$\mathcal{L}_{VAE}(\theta, \phi, x) = -\mathbb{D}_{KL}\{q_\phi(z|x)||p_\theta(z)\} + \frac{1}{L} \sum_{l=1}^{L} p_\theta(x|z), \tag{6}$$

where the first term denotes the regularisation penalty from the variational encoder, and the second term expresses decoder-induced reconstruction loss between generation and original input. Following a similar derivation, the learning objective for the EVAE can then be derived from Equation (7)),

$$\mathcal{L}_{EVAE} = \mathbb{E}_{q((z_b, z_s)|x)}\left[-\mathbb{D}_{KL}\{q((z_b, z_s)|x)||p(z_b, z_s)\} + \log p_{\theta_b, \theta_s}(\tilde{x}|(z_b, z_s))\right]. \tag{7}$$

From the previous analytical derivation, the learning objective of EVAE can be seen as comprised of two components: the first corresponds to the regularisation penalty of the derived joint encoder of our EVAE, whereas the residual term relates to the overall reconstruction errors from our derived joint decoder.

Substituting the defined joint encoder, joint decoder, and the joint prior in Equations (1), (3) and (5) into Equation (7), we can further derive the learning objective into the following form:

$$
\begin{aligned}
\mathcal{L}_{EVAE}(\theta_b, \theta_s, \phi_b, \phi_s, x) &= \mathbb{E}_{q((z_b, z_s)|x)}\Big[ -\big\{ \log(q_{\phi_b}(z_b|x)) + \log(q_{\phi_s}(z_s|x)) + \mathcal{R}_e(z_b, z_s) \\
&\quad - \log p_{\theta_b}(z_b) - \log p_{\theta_s}(z_s) \big\} + \log p_{\theta_b}(\tilde{x}|z_b) + \log p_{\theta_s}(\tilde{x}|z_s) + \mathcal{R}_d(\tilde{x_b}, \tilde{x_s}) \Big] \\
&= \mathbb{E}_{q((z_b, z_s)|x)}\Big[ -\{\log(q_{\phi_b}(z_b|x)) - \log(p_{\theta_b}(z_b))\} + \log p_{\theta_b}(\tilde{x}|z_b) \\
&\quad - \{\log(q_{\phi_s}(z_b|x)) - \log(p_{\theta_s}(z_s))\} + \log p_{\theta_s}(\tilde{x}|z_s) + [\alpha]\big\{ \mathcal{R}_e(z_b, z_s) - \mathcal{R}_d(\tilde{x_b}, \tilde{x_s}) \big\} \Big] \quad (8) \\
&= \mathbb{E}_{q((z_b, z_s)|x)}\Big[ -\mathbb{D}_{KL}\{q_{\phi_b}(z_b|x)||p(z_b)\} + \log p_{\theta_b}(\tilde{x}|z_b) - \mathbb{D}_{KL}\{q_{\phi_s}(z_s|x||p(z_s))\} \\
&\quad + \log p_{\theta_s}(\tilde{x}|z_s) + [\alpha]\big\{ \mathcal{R}_e(z_b, z_s) - \mathcal{R}_d(\tilde{x_b}, \tilde{x_s}) \big\} \Big] \\
&= \mathcal{L}_{VAE_b}(\theta_b, \phi_b, x) + \mathcal{L}_{VAE_s}(\theta_s, \phi_s, x) + [\alpha]\big\{ \mathcal{R}_e(z_b, z_s) - \mathcal{R}_d(\tilde{x_b}, \tilde{x_s}) \big\}.
\end{aligned}
$$

To rewrite the learning objective this way is to group two discrepancy functions together. Remaining terms $\mathcal{L}_{vae_b}(\theta_b, \phi_b, x)$ and $\mathcal{L}_{vae_s}(\theta_s, \phi_s, x)$ match with the learning objectives of two conventional VAEs.

### 4.3. Learning Constraints

Armed with the derived learning objective (cf. Equation (8)), a crucial follow-up question is what kind of learning constraint can be applied to this model. In short, there are two neuroscience-inspired learning constraints, i.e., dependence-minimisation and regularisation constraints, that can be applied to EVAE by simply tuning hyperparameter $\alpha$ toward its two extremes 0 and $+\infty$.

#### 4.3.1. Dependence-Minimisation Constraint

In human perceptual learning, neural representation is inefficient when its outputs are redundant. Thus, during early visual learning in human, the sensory brain of an infant employs dependence minimisation to reduce redundant neural representations [24,25]. This redundancy-reduction mechanism is harnessed to promote learning-diversified visual representations (features).

Motivated by previous findings, we applied this dependence-minimisation constraint to a deep generative model, EVAE, to encourage varied (less-redundant) generations. This dependence-minimisation constraint was introduced by setting $\alpha$ to a large number in the training EVAE. This enlarges the differences between two-component VAEs. As a result, it permits an EVAE to generate varied data instances that come from the base and scaffolding VAEs.

We hypothesise that imposing this dependence-minimisation constraint on learning EVAE allows the generation of varied data instances to improve the generative modelling performance of this deep generative model.

#### 4.3.2. Regularisation Constraint

Different to the dependence-minimisation constraint, when $\alpha$ is tuned to a small value that is close to 0, it implies a different learning constraint: the regularisation constraint. The regularisation constraint targets learning stable, interpretable representations by harnessing the external supervision signal to resolve ambiguous visual input in acquiring clear-cut visual concepts for all sorts of discriminative tasks [11,26].

To see how this learning constraint improves the discriminative modelling performance of our derived EVAE, we started by specifying a semisupervised EVAE. The only modification is on the incorporation of class label information $y$ in making the scaffolding VAE ($VAE_s$). The imposed regularisation constraint here, a relative small $\alpha$, encourages the encoded representations in an unsupervised manner (from $VAE_b$) to be regularised by discriminative information (from $VAE_s$), i.e., this regularisation process can be seen as utilising the discriminative representations that it learned from the scaffolding VAE ($VAE_s$) to regularise generative ones from the base VAE ($VAE_b$).

For this reason, we further hypothesise that imposing the regularisation constraint on learning a semisupervised EVAE allows for encoded representations to be discriminative-regularised to improve the discriminative modelling performance of such a model.

## 5. EVAE Parameter Estimation

To learn our proposed EVAE, we resorted to the reparameterisation trick in [4] to develop the optimisation algorithm. This approach directly employs Monte Carlo sampling to attain samples of latent variables.

As two sets of latent variables in EVAE are assumed to be continuous and differentiable in the real coordinates, two sets of variational parameters are approximated in mean-field and full-rank Gaussians, i.e., $z_b \sim z_b|x$ and $z_s \sim z_s|x$. Two valid reparameterisations that apply to $z_b$ and $z_s$ can be specified as $z_b = \mu_b + \sigma \cdot \epsilon_b$ and $z_s = \mu_s + L \cdot \epsilon_s$, where $\epsilon_b$ and $\epsilon_s$ are two independent noise variables that both follow normal distribution. Reconstruction errors in $\mathcal{L}_{VAE}(\theta_b, \phi_b, x)$ and $\mathcal{L}_{VAE}(\theta_s, \phi_s, x^{(i)})$ can be further expressed as $\frac{1}{L}\sum_{l=1}^{L} p_\theta(\tilde{x}|\mu_b + \sigma \cdot \epsilon_b)$ and $\frac{1}{L}\sum_{l=1}^{L} p_\theta(\tilde{x}|\mu_s + L \cdot \epsilon_s)$.

In terms of KL divergence in $\mathcal{L}_{VAE}(\theta_b, \phi_b, x^{(i)})$ and $\mathcal{L}_{VAE}(\theta_s, \phi_s, x^{(i)})$ under Gaussian approximation, their KL terms can be integrated analytically as $\frac{1}{2}\sum_{k=1}^{K}(1 + \log(\sigma_k^{(i)})^2 - (\mu_{b_k}^{(i)})^2 - (\sigma_k^{(i)})^2)$ and $\frac{1}{2}\sum_{k=1}^{K}(1 + \log(L_k^{(i)})^2 - (\mu_{s_k}^{(i)})^2 - (L_k^{(i)})^2)$ [4]. Overall maximisation objective $\mathcal{J}$ is therefore derived from Equation (9)):

$$
\begin{aligned}
\mathcal{J}_{(\theta_b, \theta_s, \phi_b, \phi_s)} = \\
\frac{1}{2}\sum_{k=1}^{K}(1 + \log(\sigma_k^{(i)})^2 - (\mu_{b_k}^{(i)})^2 - (\sigma_k^{(i)})^2) + \frac{1}{2}\sum_{k=1}^{K}(1 + \log(L_k^{(i)})^2 - (\mu_{s_k}^{(i)})^2 - (L_k^{(i)})^2) \\
+ \frac{1}{L}\sum_{l=1}^{L} p_\theta(\tilde{x}|\mu_b + \sigma \cdot \epsilon_b) + \frac{1}{L}\sum_{l=1}^{L} p_\theta(\tilde{x}|\mu_s + L \cdot \epsilon_s) + \alpha\{\mathcal{R}_e(z_s|x, z_b|x) - \mathcal{R}_d(\tilde{x}|z_b, \tilde{x}|z_s)\},
\end{aligned}
\tag{9}
$$

where $\epsilon_b \sim \mathbb{N}(0,1)$ and $\epsilon_s \sim \mathbb{N}(0,1)$. As two sets of variational parameters stay in one learning objective, we relied on alternating optimisation to iteratively derive the set of $(\theta_b, \phi_b)$ and $(\theta_b, \phi_b)$. The overall learning algorithm is summarised below as Algorithm 1.

---

**Algorithm 1** EVAE learning algorithm.

---

**Require:** Dataset $X$; encoder: $q(z_b, z_s|x; \phi_s, \phi_b)$; decoder: $p(\tilde{x}|z_b, z_s; \theta_b, \theta_b)$
  Initialise the parameters of $\theta_b; \theta_s; \phi_b; \phi_s$
  Set value of hyperparameter $[\alpha]$
  **repeat**
    **while** fix $\theta_s, \phi_s$ **do**
      random sampling from $\epsilon_b$, where $\epsilon_b \sim \mathbb{N}(0,1)$
      Approximate $\nabla_{\theta_b, \phi_b}(\mathcal{J})$ via the differentiating the learning objective in Equation (9) w.r.t
$\theta_b, \phi_b$
      Update $\theta_b, \phi_b$ via an off-shelf gradient-ascent algorithm, e.g., Adam, using $\nabla_{\theta_b, \phi_b}(\mathcal{J})$
    **end while**
    **while** fix $\theta_b, \phi_b$ **do**
      random sampling from $\epsilon_s$, where $\epsilon_s \sim \mathbb{N}(0,1)$
      Approximate $\nabla_{\theta_s, \phi_s}(\mathcal{J})$ by differentiating learning objective in Equation (9) w.r.t $\theta_s, \phi_s$
      Update $\theta_s, \phi_s$ via an off-shelf gradient-ascent algorithm, e.g., Adam, using $\nabla_{\theta_s, \phi_s}(\mathcal{J})$
    **end while**
  **until** elbo (objective) is converged to a certain level
  **Return** optimal $\theta_b; \theta_s; \phi_b; \phi_s$

---

## 6. Empirical Experiments

To test whether our introduced learning constraints were capable of improving discriminative and generative EVAE modelling performance, we deployed two types of proposed EVAE, unsupervised and semisupervised, to solve a generative and discriminative task, respectively.

Specifically, we aimed to validate the two following hypotheses:

**Hypothesis 1.** *Imposing a dependence-minimisation constraint on learning an unsupervised EVAE improves the generative modelling performance of the model, producing varied patterns of generation in a generative task.*

**Hypothesis 2.** *Imposing a regularisation constraint on learning a semisupervised EVAE improves discriminative performance in a discriminative task.*

*6.1. Experiment Setup*

For the generative task, the binarised MNIST dataset served as our training dataset; we aimed to utilise the unsupervised EVAE to generate MNIST-like handwritten digits. The goal was to observe whether the imposed dependence-minimisation constraint could lead to varied patterns of data generation between two-component VAEs in an EVAE. Thus, after the training session, both latent spaces from the base and scaffolding VAEs were harnessed for the generative task.

For the discriminative task, we chose CIFAR-10(4K) as our training dataset to classify CIFAR-10 objects. This CIFAR-10(4k) dataset is identical to the standard CIFAR-10 with all but 4000 labels only. This restricts the exploitation of labelled data to mere 4000 labels in solving a CIFAR-10 object-classification task [27]. Different to the previous generative task, before the training session, only the latent space from the scaffolding VAE (discriminative) was used for the classification task.

To speed up the training process, we further considered the usage of inference and recognition networks [23] to replace the original encoders and decoders in an EVAE. As depicted in Figure 2, this demanded the employment of four deep neural networks to wrap two sets of model and variational parameters, $\theta_b, \phi_b$ and $\theta_s, \phi_s$.

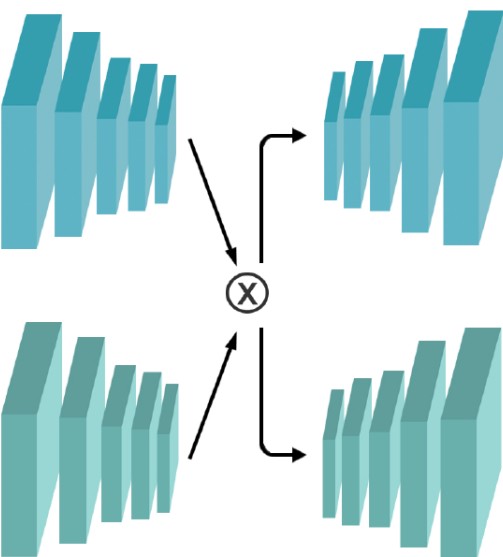

**Figure 2.** Inference and recognition networks in EVAE. In this figure, we outline usage inference and recognition networks (deep neural networks for encoders and decoders in our proposed EVAE) to wrap the model and variational parameters of the joint encoder and decoder in the EVAE. The upper panel (cyan-coloured) shows incorporated inference and recognition networks for our base VAE, whereas networks in the lower panel (green-coloured) were employed for the scaffolding VAE. Note here that $\otimes$ indicates the imposed learning constraint.

The detailed training configurations on two datasets are summarised in Table 1. Note that considering other feasible configurations for the encoders and decoders, e.g., the number of hidden

units and the type of layer wise activation, the relative performance of the model was observed as insensitive to these choices. Algorithm 1 was used to update both the model and variational parameters.

**Table 1.** Training configurations on the MNIST and CIFAR-10(4K) datasets.

| Dataset | Model Architecture and Training Configurations |
|---|---|
| MNIST digits | Input: 784 (flattened $28 \times 28 \times 1$) <br> Base encoder: FC 500, 300, ReLU activation <br> Scaffolding encoder:FC 500, 200, 100, <br> ReLU activation <br> Decoders FC 200, 784, <br> Sigmoid activation, Gaussian <br> Batch size: 100 <br> num of batch: 100 <br> optimiser: rmsprop [28] |
| CIFAR-10(4K) | Input: $32 \times 32 \times 3$ <br> Base encoder: Conv $32 \times 3 \times 3$ (stride 2), <br> Pooling(stride 2). <br> Conv $64 \times 3 \times 3$ (stride 2) FC 200. <br> ReLU activation <br> Scaffolding Encoder: Conv $32 \times 3 \times 3$ (stride 2), <br> Pooling(stride 2). <br> Conv $32 \times 3 \times 3$ (stride 2) FC 200. Categorical. <br> ReLU activation <br> Batch Size: 100 <br> num of batch: 50 <br> optimiser:Adam [29] |

### 6.2. Generative Task

To test our first hypothesis, i.e., whether the imposed dependence-minimisation constraint, a large $\alpha$, leads to varied patterns of digit generation, we trained the EVAE with the binarised MNIST dataset to generate MNIST-like digits. Under our hypothesis, a different $\alpha$ should lead to differentiated impacts on the final loss function of the EVAE. We then chose 10 representative $\alpha$ values, $0.0001, 0.001, 0.01, 0.1, 1, 5, 10, 15, 20, 25$, in the construction of ten versions of the unsupervised EVAE. For comparative purposes, we also included the case of $\alpha = 0$ to observe the generative performance of two unrelated vanilla VAEs.

To obtain an objective quantitative evaluation to assess the performance of our proposed EVAE in a generative task is difficult. Some commonly applied metrics like the estimated negative log-likelihood of a set of samples and optimised lower bound were not applicable here, as such indices are varied across the choice of $\alpha$ in an EVAE. In comparison with the fixed optimised lower bound of a conventional VAE, the optimised lower bound of an EVAE is largely varied and dependent upon the value of $\alpha$.

Since the prior-mentioned quantitative measures are less accountable, here we resorted to a qualitative evaluation to validate Hypothesis 1. The considered qualitative-evaluation metric relies on visual inspection of the generated pixel space (digits) based on the linear interpolation of the encoded latent space. This metric has widely been used in assessing the qualitative performance of a generative model [30].

For comparative purposes, we set the start and end of this linear interpolation as digits "2" and "9", respectively, to observe how the two decoders (from the base and scaffolding VAE, respectively) generate sequences of digit transformation from "2" to "9". Under Hypothesis 1, with the imposed dependence-minimisation constraint, that is, a large $\alpha$, the generative processes of two component VAEs diverged, leading toward varied patterns of digit generation for this generative task.

As shown in Figure 3, with small $\alpha$ values (0.1 and 1), the generative processes of two-component VAEs (base and scaffolding VAEs) in EVAE were unfolded in an identical manner. Sequences of images on digit transformation from two-component VAEs did not differ in terms of visual inspection. Both generative processes underwent the transition from digit "2" to "1", then finally toward digit "9".

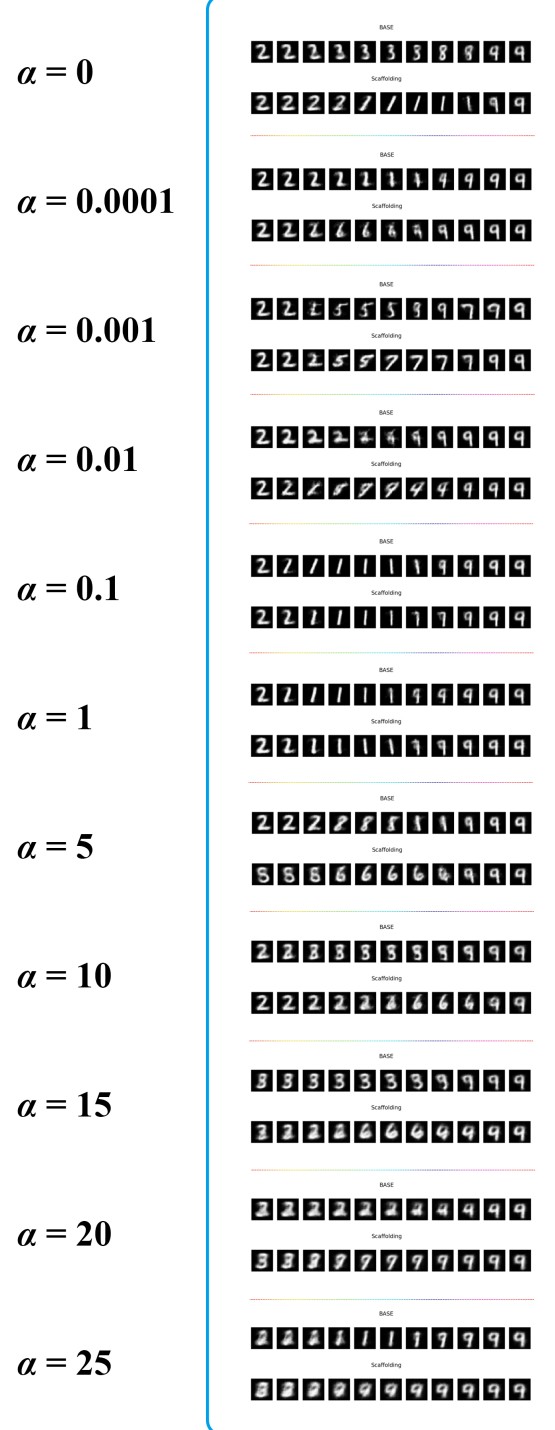

**Figure 3.** $\alpha$-wise linear interpolation of latent spaces of base and scaffolding VAEs in EVAE on MNIST dataset. For comparative purposes, we fixed the start and end digits to be "2" and "9", respectively. For each $\alpha$, we linearly interpolate the latent spaces of the two-component VAEs in EVAE to observe their entire digit-transformation process.

With the imposed dependence-minimisation constraint, that is, the higher $\alpha$ values (5, 10, and 15), the generative processes of two-component VAEs are exhibited in two different ways. $\alpha = 10$ serving as an example, the base VAE unfolds the generation process through digit "8", whereas the scaffolding VAE unfolds its generation through digit "6". Moreover, besides the observed differentiated generative processes of two-component VAEs in EVAE, with further elevated $\alpha$ values (20 and 25), both generative processes of two-component VAEs are heavily influenced by the start and end signal (digits). Both generative processes focused on producing start digit "2" and end digit "9" in various scales and orientations. However, with an elevated $\alpha$, the generated digits appeared to be obscure and less crisp in comparison with those small $\alpha$ cases (0.1 and 1). For comparative purposes, we also included the $\alpha = 0$ case to degrade EVAE to two independent VAEs. Reflecting on the actual generation processes of two VAEs, two linearly interpolated digits were drastically different.

Unfortunately, further increment on $\alpha$ leads to a case of collapsed generation. A demonstration of this collapsed generation is given in Appendix A.

Armed with these generated (interpolated) sequences of images on digit transformation, we ran a small-scale behaviour experiment, inviting 10 individuals to assess and rate the similarity of generated digits between base and scaffolding VAEs. All 10 raters were blind to the value of $\alpha$ to give subjective similarity ratings on two linearly interpolated sequences of images on generated digits independently for each $\alpha$. The rating statistics were collected with a five-level Likert scale, with "1" indicating the most similar generation pattern.

The $\alpha$-wise similarity rating statistics are reported in Table 2. From Table 2, we observe the trend that, with an elevated $\alpha$, generated digits from two-component VAEs are more likely to be judged as different generations. As $\alpha$ was set to either 0.1 or 1, the generated digits from two-component VAEs were judged to be nearly identical with each other. We further ran a one-way analysis-of-variance (ANOVA) test to assess whether these $\alpha$-wise similarity ratings differed statistically. The computed F-statistic was 15.31 ($p < 0.001$), suggesting that imposing the dependence-minimisation constraint, that is, tuning $\alpha$ to a large value on EVAE, ensures differentiated generative processes for two-component VAEs in EVAE, which improves the generative modelling performance of an EVAE.

**Table 2.** $\alpha$-wise similarity ratings on linearly interpolated digits from two-component VAEs in EVAE.

| the value of $\alpha$ | 0 | 0.0001 | 0.001 | 0.01 | 0.1 | 1 | 5 | 10 | 15 | 20 | 25 |
|---|---|---|---|---|---|---|---|---|---|---|---|
| average similarity rating | 2.5 | 2.6 | 2.9 | 2.6 | 1.2 | 1.1 | 3.7 | 2.9 | 3.2 | 2.9 | 3.3 |

*6.3. Discriminative Task*

To validate our second hypothesis, i.e., whether the regularisation learning constraint, a lower $\alpha$, leads to a better-performing semisupervised EVAE in solving a classification task, we constructed 11 versions of a semisupervised EVAE with the same model architecture but varied $\alpha$, that is, $0, 0.0001, 0.001, 0.01, 0.1, 1, 5, 10, 15, 20, 25$. To be consistent with unsupervised EVAE models in the above generative task, we adopted an identical $\alpha$ to construct these 11 versions of semisupervised EVAE for this discriminative task.

For comparative purposes, we pitted these 11 semisupervised EVAE models against existing state-of-the-art semisupervised learning approaches on the CIFAR-10(4k) dataset. Importantly, to avoid a conflating comparison, rather than reporting the classification performance of these state-of-the-art semisupervised learning approaches that were implemented in previous comparative studies [27,31], we also implemented some of these learning approaches on our testbed. These state-of-the-art learning approaches went through an identical training curriculum as the derived semisupervised EVAE.

These state-of-the-art models range from the ladder network [19], the spike-and-slab sparse-coding approach (S3C) [20], to the virtual adversarial training approach (VAT) with $\epsilon = 1.0 \& \zeta = 1e^{-4}$ [21]), the mean-teacher learning approach [32] and the Speed-as-a-Supervisor (SaaS) learning algorithm [31]. Lastly, to serve as the benchmark, we also included a purely supervised

approach, i.e., a Bayesian neural network (BNN), which was trained on 4000 annotated CIFAR-10 training samples in this experiment. For more reliable results, we ran each candidate model five times to report its average performance (with standard deviation) on the CIFAR-10(4K) classification task in Table 3.

**Table 3.** Test error rates on CIFAR-10(4k) dataset. Note: S3C, spike-and-slab sparse coding; VAT, virtual adversarial training; SaaS, Speed as a Supervisor; BNN, Bayesian neural network.

| Model | Test Error(%) |
|---|---|
| **Our implementation** | |
| S3C [20] | $28.1 \pm 0.3$ |
| Ladder networks [19] | $16.5 \pm 0.3$ |
| VAT [21] | $24.1 \pm 1.2$ |
| semisupervised EVAE ($\alpha = 25$) | $47.6 \pm 0.6$ |
| semisupervised EVAE ($\alpha = 20$) | $48.3 \pm 0.1$ |
| semisupervised EVAE ($\alpha = 15$) | $47.4 \pm 0.4$ |
| semisupervised EVAE ($\alpha = 10$) | $47.6 \pm 0.4$ |
| semisupervised EVAE ($\alpha = 5$) | $47.8 \pm 0.3$ |
| semisupervised EVAE ($\alpha = 1$) | $46.4 \pm 0.2$ |
| semisupervised EVAE ($\alpha = 0.1$) | $21.7 \pm 0.4$ |
| semisupervised EVAE ($\alpha = 0.01$) | $19.5 \pm 0.2$ |
| *semisupervised EVAE ($\alpha = 0.001$)* | $18.2 \pm 0.1$ |
| semisupervised EVAE ($\alpha = 0.0001$) | $18.3 \pm 0.1$ |
| semisupervised EVAE ($\alpha = 0$) | $22.5 \pm 0.3$ |
| Benchmark: supervised BNN | $24.1 \pm 0.2$ |
| **Other implementations** | |
| VAT [21] (in [27]) | $13.13 \pm 0.3$ |
| VAT + EntMin [21,33] (in [31]) | $10.55$ |
| Mean Teacher [32] (in [27]) | $15.87 \pm 0.3$ |
| Mean Teacher [32] (in [31]) | $12.31$ |
| SaaS [31] (in [31]) | $10.94 \pm 0.1$ |

From the reported test error rates in Table 3, it is clear that, under our implementation, except for the fine-tuned ladder networks [19] (test error rate at $16.5 \pm 0.3$), imposing the regularisation learning constraint on a semisupervised EVAE, ($\alpha = 0.001$ (test error rate at $18.2 \pm 0.1$)) could produce a competitive classification result even in the face of state-of-the-art semisupervised learning algorithms.

However, from the bottom part of Table 3, even the best-performing semisupervised EVAE ($\alpha = 0.001$) was inferior to state-of-the-art semisupervised learning approaches under other implementations. This observed performance gap may be due to different adopted training curricula and the less-convoluted base model we used in our implementation.

Diving deeper into our implemented 11 versions of a semisupervised EVAE, as demonstrated in Figure 4, the empirical results of these 11 semisupervised EVAE variants on the classification benchmark reveal an interesting relation between the $\alpha$ value and the final test error rate. With the imposed regularisation constraint, that is, the small $\alpha$ values, their corresponding semisupervised EVAE models performed better compared to large $\alpha$ value alternatives in terms of classification test error rates ($18.2\%(\alpha = 0.001)$ vs. $47.6\%(\alpha = 25)$). Surprisingly, with a further-minimised $\alpha = 0.0001$, no greater performance gain was observed in this experiment. In the extreme case of $\alpha = 0$, its corresponding semisupervised EVAE also lost its competitive edge with the cited state-of-the-art learning approaches. Another interesting finding is that as we gradually pushed the $\alpha$ value to its positive end, the classification performance of the semisupervised EVAE deteriorated dramatically: $46.4\%(\alpha = 1)$.

These compelling empirical results suggest that, with an imposed regularisation constraint of a small $\alpha$, we can improve the discriminative modelling performance of this derived semisupervised EVAE on a classification task.

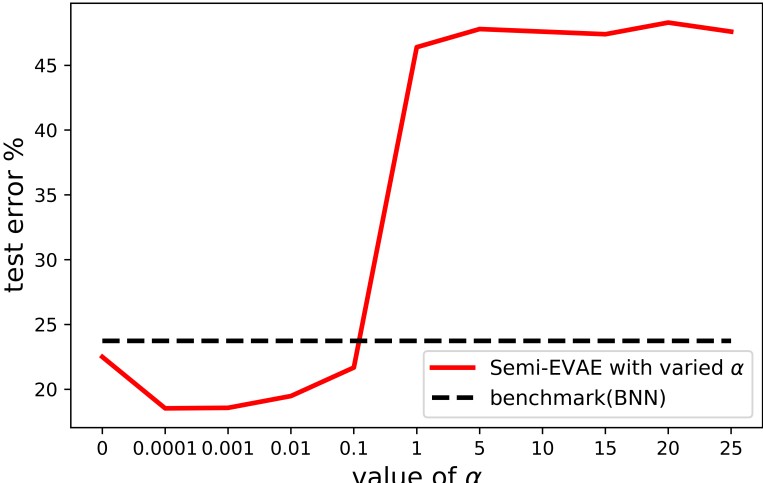

**Figure 4.** Effect of $\alpha$ on discriminative performance of semisupervised EVAE. This figure demonstrates the classification performance of 11 versions of a semisupervised EVAE on the CIFAR-10(4K) dataset. Note that the benchmark model (a fully supervised Bayesian neural network) was implemented in a similar model architecture as the scaffolding encoder that is detailed in Table 1.

## 7. Conclusions

In this research, we introduced a novel form of variational autoencoder, i.e., EVAEs, to stack two variational autoencoders in a scaffold. Imposing neuroscience-inspired dependence minimisation and regularisation constraints on the derived EVAE, we were able to improve both the generative and discriminative modelling performance of this deep generative model. The imposed dependence-minimisation constraint allows the EVAE to generate diversified patterns of generation. Running this constrained EVAE on the MNIST dataset allowed the model to generate MNIST-like digits in varied patterns. The imposed regularisation constraint, in contrast, enforced encoded representations from two-component VAEs in EVAE to be coincided with each other. On the CIFAR-10(4K) dataset, implementation of this regularisation constraint improved the discriminative performance of a semisupervised EVAE to achieve competitive empirical results, even in the face of state-of-the-art semisupervised learning approaches.

However, there are several remaining issues that need to be addressed in future studies. The primary one is to seek a theoretical approach to derive a precise range of $\alpha$ to prevent the occurrence of collapsed generation that is shown in Appendix A. Moreover, implementing these learning constraints on a deep generative model also comes with the price of prolonged training. We believe that more careful tweaking of the model architecture could close this gap.

**Author Contributions:** W.B. and C.Q. conceived and designed the model; W.B. performed the experiment and analysed the results; W.B. wrote the preliminary version of this manuscript; and Z.-W.L. and C.Q. revised the manuscript.

**Funding:** This study was partially supported by the National Natural Foundation of China under Grant No. 61472117.

**Conflicts of Interest:** The authors declare no conflict of interest.

**Abbreviations**

The following abbreviations are used in this manuscript:

| | |
|---|---|
| VAE | Variational Autoencoder |
| EVAE | Encapsulated Variational Autoencoders |
| BNN | Bayesian Neural Network |
| ReLU | Rectified Linear Unit |
| ADVI | Automatic Differentiation Variational Inference |
| S3C | Spike-and-Slab Sparse-Coding Approach |
| VAT | Virtual Adversarial Training |
| FC | Fully Connected Layer |
| Conv | Convolutional Layer |
| ANOVA | Analysis of Variance |
| SaaS | Speed as a Supervisor |

**Appendix A. Collapsed Generation**

Intriguingly, if $\alpha$ was set to a relatively large rate, e.g., 200 or higher, the generative performance of our proposed EVAE was largely attenuated, leading to the devastating collapsed generation. Demonstrated in Figure A1, the generated images were meaningless and had limited relation with the MNIST-like digits. Preventing the occurrence of this collapsed generation calls for a theoretical approach to define a more precise and practical range of $\alpha$ in future studies.

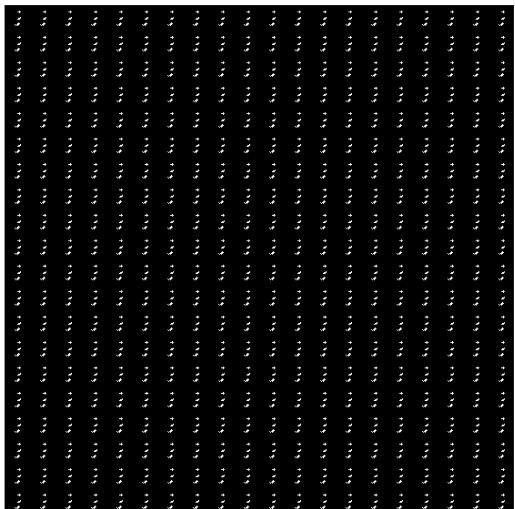

**Figure A1.** Collapsed generation.

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
