# Peer review of "Improving Generative and Discriminative Modelling Performance by Implementing Learning Constraints in Encapsulated Variational Autoencoders"

_applsci, doi:10.3390/app9122551_

Round 1
Reviewer 1 Report
This paper aims to improve the modelling performance of a deep generative model in both generative and discriminative tasks via introducing two neuroscience inspired learning constraints. To turn this idea into a concrete theoretical model, authors introduce a variant of variational auto-encoder: EVAE, then adding a learning associated hyper-parameter to implement authors mentioned learning constraints. Authors’ theoretical efforts and empirical validation on this issue should be recognised. However, there are several places are not clear and revisions on these places are certainly needed for further consideration. I use point-wise format to state these unclear places.
(1)In Introduction 1.1., a brief history review on deep generative model should be added in this section. Moreover, since authors insist the learning constraints are inspired from neuroscience, care to explain why to introduce these neuroscience constraints at first place? Or are there any pre-existing works that also attempt to build biological/neurological compatible generative model?
(2)In Related work section, a more comprehensive review on recent advances of semi-supervised learning algorithms should be provided here.
(3)In the methodology section, it will be more readable if authors can provide a tabular format of comparison between VAE and the introduced EVAE.
(4)Authors define the hyper-parameter alpha to be semi-positive, care to explain why?
(5)The main of the introduced EVAE stands on using two VAEs instead of one to probe their relations to see such relations have implications on representation learning. Such idea seems similar to Sara et al 2017 work on capsule network and Hinton et al 2014 early work on Transforming auto-encoders. Can authors explain the differences between your model and theirs?
(6)In section on empirical validation, the content on two experiments (on discriminative and generative tasks) is not detailed unfolded. More detailed description on how to implement EVAE in two tasks should be provided here.
(7)The size of Figure 4 should be enlarged as it demonstrates key findings of EVAE on generative task.
(8)As authors pit semi-supervised EVAE with other mainstream semi-supervised learning models, each model configuration on benchmark dataset should be rendered in a more clear and thorough manner, rather than several lines in this manuscript.
(9)The professional English check is required as this manuscript appears to have several unclear expressions and grammatical issues.
I wish authors can spare time to revise the foregoing points and clear the pertaining confusion.
Author Response
Please find the response letter attached below.

Reviewer 2 Report
The authors propose a novel VAE based architecture (encapsulated VAE) and claim that such architecture improves generative and discriminative tasks. The proposed architecture is novel and interesting, however the validation is currently weak for two reasons: 1) In terms of generative power, the proposed method is not compared to any existing method. Even in the best shown case, non-realistic numbers are produced. So with the evidence provided, I can't say it is a good generative method. 2) In terms of classification power, the sensitivity analysis with alpha needs to be extended to values of alpha over a wider range, especially to lower values of alpha to check if it ever beats the cascaded networks.
From a methodology perspective, two hypothesis are formulated, but not enough evidence to validate them is provided.
Last, I think the paper should be proof-read for clarity.
# Major concerns and comments
1. Role and values of alpha
Authors claim that making alpha very large, e.g. alpha -> infinity, enforces statistical independence of z_s and z_b. However, making alpha->infinity will neglect the loss of each VAE, hence the loss considers equations (2) and (4). Specifically, (2) decreases as zb is different from zs. This enforces that zs and zs are *different* but I am not sure why "statistically independent". Please discuss and justify the relation between different and independent representations.
If alpha=0, then the cost function does not enforce any relation between the two VAEs. It is not clear how then the VAEs can help regularise the VAEb.
Moreover, in the experiments, alpha=0.3, 0.5, 50, 100. To support the claims about the extreme cases, a much smaller (e.g. 0) and a much higher (e.g. 1E10) alpha values should be checked. At any rate, more values need to be tested to draw conclusions about performance w.r.t. alpha.
2. Experimental set-up
With the proposed EVAE, two latent spaces are learnt, and two reconstructed images are obtained. I am confused which (or if both, how) of the latent representations is used for the classification task.
3. Validation
Qualitative results are confusing. Fig. 3 must be further explained: what should the reader focus the attention on? For each value of alpha, the reader would expect a transition between the same two digits, for example from 9 to 2. Otherwise how can a qualitative analysis be made if each alpha shows different things? I'd suggest that authors prove generative power by interpolating, in the latent space, from two locations corresponding to the latent representation of two input images, so they can always pick the same. Also, if authors want to make a qualitative analysis, doing so based on 5 figures is not enough. They should generate 10s of images and have different human raters assess them, blind to the value of alpha and to whether they were generated by the VAEb or the VAEs.
The latent representations (2D scatter plot) are remarkably similar between the VAEb and the VAEs (except for low alpha values). This further suggest my comment above and raises concerns about the ability of VAEs to regularise VAEb with alpha = 0.
Also, it is difficult to draw any conclusions without comparing to, at least, VAE on its own (e.g. the base VAE trained without the scaffolding one).
In table 2, if decreasing alpha improved the results, why did authors not decrease it further, and see when does the improvement stop? I'd particularly like to see the result for alpha =0, and compare it to a standard VAE. Same with Fig. 5.
# Minor remarks
I think the title should be rewritten, to incorporate the main contribution (encapsulated variational auto-encoders -EVAE).
Figure 1 is a bit misleading, since it makes the reader feel that the two decoded representations are somehow fused into x. This is clearly not true (from Fig 2). Please amend or clarify/
Please revise the equations and the notation. It is a bit unclear how equations (1) and (2) relate: if Re(zb,zs) is an exponential, then we have the exponential of an exponential in (1), which does not seem to make sense.
Fig. 3 seems to have 2 captions. Please unify into one.
p5-l148 "in practise a single sample suffice". Why?
p11-l289 " We ran each model five times, then averaged the performance." Why would this make sense? Does this mean that the training did not converge? This might suggest problems with the optimization parameters (learning rate, number of iterations, etc).
# Typos / wording
p1-l12: EVAE -> define the acronym here (it is only defined afterwards)
p1-l14: with its learning -> with their learning
p2-l32: GAN, is difficulty ->GAN is difficult
p3-l94: unobservable(latent) -> space missing before the parenthesis.
"chiefly" is used throughout, liberally. It should be restricted to the main points.
Author Response
We appreciate the reviewer for sparing time to read our manuscript, the detailed response letter can be found below as an attachment.

Round 2
Reviewer 1 Report
The authors have addressed my previous comments, as such, I'd like to recommend an Accept.
BTW, please double check the reference formats, making them consistent, say [7].
Author Response
please see the attached word document.

Reviewer 2 Report
The authors have addressed my previous comments partially. They did answer satisfactorily my doubts about experimental set-up, only partly my doubts about qualitative analysis of the generative model, and important improvenets validation and clarity are still needed:
1. Clarity:
I raised lack of clarity as one of my major concerns, requesting a proof-read. I think there are quite a few sentences that still do not read well, and that need to be written correctly. Just some examples:
- Armed with our proposed EVAE, the dependence minimisation learning constraint can be integrated in learning of EVAE, it largely improves the generative modelling performance of EVAE.
- The former approach targets on incorporating the supervisions of some kind
2. Evaluation/role and values of alpha:
In my previous review, I stated as a major concern that "Otherwise how can a qualitative analysis be made if each alpha shows different things? I'd suggest that authors prove generative power by interpolating, in the latent space, from two locations corresponding to the latent representation of two input images, so they can always pick the same. Also, if authors want to make a qualitative analysis, doing so based on 5 figures is not enough. They should generate 10s of images and have different human raters assess them, blind to the value of alpha and to whether they were generated by the VAEb or the VAEs." The authors have partly answered this by running "a small scale behaviour experiment to collect 10 human ratings on the similarity of generated digits between the base and scaffolding VAEs. All human raters are blind to the value of alpha, and the rating statistics are collected in a five point Likert scale manner (1 to 5, 1 indicates the least similar generation patterns)". However we need more information: how many images did the humans rate; what were they comparing (10x10 figures as in Fig. 3? or individual digits?); what is the variability between users; is there any statistical significance on the difference in average ratings?
Moreover, their comments on these results (Fig 3) are quite limited. For example:
"As shown in Figure 3, with small α values (0.1, 0.5, 1), the generated digits between two component VAEs in EVAE are rendered in a highly similar manner."
However, the latent space scatter plots look different, and we do not know where from this latent space the digit figures were generated.
In my previous review, I suggested that they take samples from the test set and interpolate. Let me clarify this further with an example. Let's pick four samples from the test set: a '4' digit , a '7' , a '6' digit and a '0' digit. The corresponding position on the latent spaces can be retrieved by using the encoders in all models, thus finding the same corresponding latent position for both scaffolding and base VAEs and for all values of alpha. Now, put these 4 digits at the corners of a 10x10 image (like the ones of Fig 3) and interpolate linearly on each latent space. This should give a figure similar to Fig. 3, but where all cases are comparable. I imaginge that other options would also be valid, but the current one is too difficult to interpret to extract conclusions.
Moreover, my suggestion of comparing to a vanilla VAE has been ignored. How can we assess (qualitatively) if the proposed architecture benefits from the joint learning of the two VAEs without comparing to what would happen with only one? This should correspond to using alpha=0. Authors say in their answer to my previous review that "we attach the ‘0’ case in the Appendix section", however in the appendix there is only the case "if α is set to a relative large rate, e.g., 200 or higher". Please do include the alph=0 results, either in Fig 3 (preferably) or in the appendix.
They showed results for alpha = 0.1, 0.5, 1, 5, 10, 15, 20, 25, 30, 40 for the qualitative experiments, but alpha 0.0001, 0.001, 0.01, 0.1 and 1 for the classification experiments. Why? This looks very ad-hoc. The same range of alpha should be used for both tasks. Also, representing the x axis in Fig 4 in decreasing order looks strange.
Author Response
Please consult the attached document below.
